# Parameterization of Size of Organic and Secondary Inorganic Aerosol for Efficient Representation of Global Aerosol Optical Properties

Haihui Zhu[1]*, Randall V. Martin[1,2], Betty Croft[2,1], Shixian Zhai[3], Chi Li[1], Liam Bindle[1], Jeffrey R. Pierce[4], Rachel Y.-W. Chang[2], Bruce E. Anderson[5], Luke D. Ziemba[5], Johnathan W. Hair[5], Richard A. Ferrare[5], Chris A. Hostetler[5], Inderjeet Singh[1], Deepangsu Chatterjee[1], Jose L. Jimenez[6], Pedro Campuzano-Jost[6], Benjamin A. Nault[7], Jack E. Dibb[8], Joshua S. Schwarz[9], Andrew Weinheimer[10]

[1] Department of Energy, Environmental & Chemical Engineering, Washington University in St. Louis, St. Louis, MO, USA

[2] Department of Physics and Atmospheric Science, Dalhousie University, Halifax, Nova Scotia, Canada

[3] Harvard John A. Paulson School of Engineering and Applied Sciences, Harvard University, Cambridge, MA, USA

[4] Department of Atmospheric Science, Colorado State University, Fort Collins, CO, USA

[5] NASA Langley Research Center, Hampton, VA, USA

[6] Cooperative Institute for Research in Environmental Sciences and Department of Chemistry, University of Colorado, Boulder, CO, USA

[7] Center for Aerosol and Cloud Chemistry, Aerodyne Research, Inc., Billerica, MA, USA

[8] Institute for the Study of Earth, Oceans, and Space, University of New Hampshire, Durham, NH, USA

[9] National Oceanic and Atmospheric Administration Chemical Sciences Laboratory, Boulder, CO, USA

[10] National Center for Atmospheric Research, Boulder, CO, USA

*Correspondence*: Haihui Zhu (haihuizhu@wustl.edu)

**Abstract** Accurate representation of aerosol optical properties is essential for modeling and remote sensing of atmospheric aerosols. Although aerosol optical properties are strongly dependent upon the aerosol size distribution, use of detailed aerosol microphysics schemes in global atmospheric models is inhibited by associated computational demands. Computationally efficient parameterizations for aerosol size are needed. In this study, airborne measurements over the United States (DISCOVER-AQ) and South Korea (KORUS-AQ) are interpreted with a global chemical transport model (GEOS-Chem) to investigate the variation in aerosol size when organic matter (OM) and sulfate-nitrate-ammonium (SNA) are the dominant aerosol components. The airborne measurements exhibit a strong correlation ($r = 0.83$) between dry aerosol size and the sum of OM and SNA mass concentration ($M_{SNAOM}$). A global microphysical simulation (GEOS-Chem-TOMAS) indicates that $M_{SNAOM}$, and the ratio between the two components ($\frac{OM}{SNA}$) are the major indicators for SNA and OM dry aerosol size. A parameterization of dry effective radius ($R_{eff}$) for SNA and OM aerosol is designed to represent the airborne measurements ($R^2 = 0.74$, slope = 1.00) and the GEOS-Chem-TOMAS simulation ($R^2 = 0.72$, slope = 0.81). When applied in the GEOS-Chem high-performance model, this parameterization improves the agreement between the simulated aerosol optical depth (AOD) and the ground-

measured AOD from the Aerosol Robotic Network (AERONET; $R^2$ from 0.68 to 0.73, slope from 0.75 to 0.96). Thus,
this parameterization offers a computationally efficient method to represent aerosol size dynamically.

## 1    Introduction

Aerosol size has numerous effects on aerosol physical and chemical properties and further on atmospheric chemistry.
Aerosol size-dependent heterogeneous chemistry affects gaseous oxidants that in turn affect production rates of
aerosol components such as sulfate and secondary organic aerosol (Ervens et al., 2011; Estillore et al., 2016). Aerosol
size also affects loss rates due to dry and wet deposition (Seinfeld and Pandis, 2016). Both direct and indirect aerosol
radiative forcing are sensitive to aerosol size, as aerosol size affects the interaction between particles and radiation,
and the rate at which a particle grows to a cloud droplet (Adams and Seinfeld, 2002; Faxvog and Roessler, 1978;
Mishchenko et al., 2002; Emerson et al., 2020). The size dependence of aerosol extinction and scattering phase
function also affects the retrieval of aerosol properties from satellites (Levy et al., 2013; Kahn et al., 2005; Jin et al.,
2023). Aerosol size affects the fraction of particles that deposit in the body when breathing as well as location within
the body where they deposit (Hinds and Zhu, 1999). An appropriate representation of aerosol size is essential for
modeling aerosol composition and optical properties (Kodros and Pierce, 2017), interpreting satellite data (Levy et al.,
2013; Kahn et al., 2005), studying climate processes (Twomey, 2007; Kellogg, 1980), and moving from aerosol
exposure towards dose in health studies (Kodros et al., 2018).
The evolution of the aerosol size distribution is affected by various processes, such as nucleation, condensation,
coagulation, and deposition. Nucleation events contribute to the number of particles in the nucleation mode (diameters
less than about 10 nm) and thus tend to decrease the mean aerosol size for a population (Aalto et al., 2001). In polluted
areas with high emission rates of aerosol precursors, mean aerosol size tends to increase by condensation and
coagulation (Sakamoto et al., 2016; Sun et al., 2011). Dry and wet aerosol deposition have strong size dependencies
due to competing physical processes (Emerson et al., 2020; Ruijrok et al., 1995; Reutter et al., 2009). The aerosol size
distribution can be simulated using aerosol microphysical schemes, such as the TwO Moment Aerosol Sectional
(TOMAS; Adams and Seinfeld, 2002) microphysics model, the Advanced Particle Microphysics (APM; Yu and Luo,
2009) model, the Global Model of Aerosol Processes (GLOMAP; Mann et al., 2010), and the Modal Aerosol Module
(MAM4; Liu et al., 2016). These schemes have valuable prognostic capabilities; however, their computational cost
has limited their use in Chemistry Climate Models (CCMs) or Chemical Transport Models (CTMs). For example, the
wall clock time increases by about 2.5 times when APM is enabled in GEOS-Chem CTM relative to the bulk model
(GCST et al., 2023). Only 3 of the 10 models that included aerosols, studied by the Atmospheric Chemistry and
Climate Model Intercomparison Project, include online size-resolved aerosol microphysics, reflecting its
computational cost and complexity (Lamarque et al., 2013; Liu et al., 2012; Szopa et al., 2013; Kodros and Pierce,

65  2017).

Methods are needed to better represent aerosol size in CCMs or CTMs without a microphysics scheme (referred to as
bulk models). These bulk models usually use prescribed relationships to obtain size-resolved aerosol properties (Croft
et al., 2005; Karydis et al., 2011; Zhai et al., 2021), which may insufficiently represent the temporal and spatial
variation (Kodros and Pierce, 2017). For example, in the GEOS-Chem CTM, a fixed dry aerosol geometric mean
radius ($R_g$) is assumed for organic matter (OM) and sulfate-nitrate-ammonium (SNA), which is based on analysis of
long-term aerosol composition and scattering measurements provided by the IMPROVE network across the
continental U.S. (Latimer and Martin, 2019). However, subsequent analysis by Zhai *et al.* (2021) found that this
aerosol size underestimated the aerosol mass scattering efficiency and the aerosol extinction coefficients during an
aircraft campaign over South Korea (KORUS-AQ). Thus, neglect of aerosol microphysical processes that shape
aerosol size distributions can be a significant source of uncertainty in aerosol optical properties in a CTM. A balance
between computational cost and representativeness of aerosol size is needed. One option is to use models with size-
resolved aerosol microphysics models to inform bulk models, such as was done for the parameterization of biomass
burning aerosol size by Sakamoto *et al.* (2016).
Recent airborne measurements offer information to evaluate and improve the simulation of aerosol size. DISCOVER-
AQ (Deriving Information on Surface Conditions from Column and Vertically Resolved Observations Relevant to Air
Quality) was a multi-year campaign over four U.S. cities that provides 3-D resolved measurements of atmospheric
gas composition, aerosol composition, size distribution, and optical properties (Choi et al., 2020; Sawamura et al.,
2017; Chu et al., 2015). KORUS-AQ (the Korea-United States Air Quality Study) offers similar measurements in a
different environment with higher aerosol mass loadings (Choi et al., 2020; Zhai et al., 2021; Nault et al., 2018; Jordan
et al., 2020).
To study the global variation in aerosol size, we explore airborne measurements from DISCOVER-AQ and KORUS-
AQ, as well as output from the GEOS-Chem-TOMAS microphysics model. We focus on OM and SNA, which
dominate fine aerosol composition in populated areas (Weagle et al., 2018; Geng et al., 2017; Meng et al., 2019; Van
Donkelaar et al., 2019; Li et al., 2017). The driving factors for variation in aerosol size are examined. A
parameterization of aerosol size using these driving factors is proposed. This parameterization is then applied to a
GEOS-Chem high-performance model bulk simulation for global aerosol optical depth (AOD), which is evaluated by
ground-measured AOD from the Aerosol Robotic Network (AERONET).
**2    Observations and Models**
**2.1    Observations**
**2.1.1    Aircraft measurements**
We examine airborne measurements from two NASA campaigns, DISCOVER-AQ and KORUS-AQ. DISCOVER-
AQ includes four deployments in Maryland (MD), California (CA), Texas (TX), and Colorado (CO). KORUS-AQ is
an international cooperative field study program conducted in South Korea (KO), sponsored by NASA and the South
Korean government through the National Institute of Environmental Research. The year as well as the date and altitude
ranges of each deployment are in Table 1.

**Table 1. Temporal and spatial coverage of each aircraft deployment**

| Campaign | Year | Date Range | Altitude from surface |
|----------|------|------------|----------------------|
| MD | 2011 | 07/01-07/29 | 0 to 5 km |
| TX | 2013 | 09/04-09/29 | 0 to 5 km |
| CA | 2013 | 01/16-02/06 | 0 to 4 km |
| CO | 2014 | 07/17-08/10 | 0 to 6 km |
| KO | 2016 | 05/02-06/11 | 0 to 8 km |

Measurements used in this study include aerosol composition, ambient aerosol extinction, aerosol number size distribution, gas tracer species, and meteorological data. Measurement methods are listed in Table 2. Measured aerosol mass is converted from standard to ambient condition before analysis using ambient temperature and pressure. We use OM directly measured during KORUS-AQ. We use water soluble organic carbon (OC) and a parameterized ratio between OM and OC (Philip et al., 2014) to calculate OM for DISCOVER-AQ. The parameterized OM is evaluated with KORUS-AQ data, and overall consistency is found (Figure A1; Appendix A). For both campaigns, dust concentration is derived from $Ca^{2+}$ and $Na^+$ assuming non-sea salt $Ca^{2+}$ accounts for 7.1% of dust mass (Shah et al., 2020):

$$Dust = \frac{\left([Ca^{2+}] - 0.0439\frac{[Na^+]}{2}\right)}{0.071} \qquad \text{Eqn. (1)}$$

Sea salt is calculated from measured $Na^+$ following previous studies (Remoundaki et al., 2013; Malm et al., 1994; Snider et al., 2016). The crustal component is removed by subtracting 10 % of $[Al^{3+}]$ (Remoundaki et al., 2013). A 2.54 scalar is applied to $[Na^+]_{ss}$ to account for $[Cl^-]$ (Malm et al., 1994):

$$Sea\ Salt = 2.54([Na^+] - 0.1[Al^{3+}]) \qquad \text{Eqn. (2)}$$

Effective radius ($R_{eff}$) (Hansen and Travis, 1974), defined as the area-weighted mean radius of a particle population, is used as a surrogate for aerosol size:

$$R_{eff} = \frac{\int r\pi r^2 n(r)dr}{\int \pi r^2 n(r)dr} \qquad \text{Eqn. (3)}$$

Measurement data are screened for dust influence by excluding data with the sum of SNA and OM ($M_{SNAOM}$) < 4 × dust mass.

**Table 2. Aircraft observations used in this study**[*]

| Variables | DISCOVER-AQ | KORUS-AQ |
|---|---|---|
| Bulk aerosol ionic composition | IC-PILS [a] | SAGA [b] |
| Sub-micron non-refractory aerosol composition | TOC-PILS [c] | HR-ToF-AMS [d] |
| Refractory black carbon concentration | SP2 [e] | |
| Dry aerosol size distribution | UHSAS [f] or LAS [g] | LAS [g] |
| Aerosol extinction profile at 532 nm | HSRL [h] | |
| NO$_2$ | 4-Channel Chemiluminescence Instrument [i] | |
| Relative humidity (RH) | DLH [j] | |

[*] Adapted from Zhai et al. (2021)
[a] Ion Chromatography Particle-Into-Liquid Sampler, with a 1.3 μm inlet cutoff aerodynamic diameter (Lee et al., 2003;
Hayes et al., 2013).
[b] Soluble Acidic Gases and Aerosol (SAGA) instrument (Dibb et al., 2003). The cutoff aerodynamic diameter of the
inlet is around 4 μm (McNaughton et al., 2007).
[c] Water-soluble organic carbon Particle-Into-Liquid Sampler, with a 1 μm inlet cutoff diameter at 1 atmosphere
ambient pressure (Sullivan et al., 2019; Timonen et al., 2010).
[d] University of Colorado Boulder High-Resolution Time-of-Flight Aerosol Mass Spectrometer (HR-ToF-AMS) with
a 1 μm inlet cutoff diameter (Nault et al., 2018; Guo et al., 2021; Canagaratna et al., 2007).
[e] Single-Particle Soot Photometer (SP2), measuring refractory black carbon with a volume equivalent diameter of 100-
500 nm (Lamb et al., 2018; Schwarz et al., 2006).
[f] Particles with mobility diameters between 60 to 1000 nm can be measured by Ultra-High Sensitivity Aerosol
Spectrometer (UHSAS), which illuminates particles with a laser and relate the single-particle light scattering intensity
and count rate measured over a wide range of angles to the size-dependent particle concentration (Moore et al., 2021).
Particles in the sample are dried to less than 20 % RH.
[g] Particles between 100 to 5000 nm measured by Laser Aerosol Spectrometer (LAS, TSI model 3340). The principle
of LAS is the same as that of UHSAS, but with a different laser wavelength (1054 nm for the UHSAS and 633 nm for
the LAS) and intensity (about 100 times higher for the UHSAS). These differences affect how the instrument sizes
non-spherical or absorbing aerosols (Moore et al., 2021). Particles in the sample are dried to less than 20 % RH.
[h] NASA Langley airborne High Spectral Resolution Lidar (HSRL) (Hair et al., 2008).
[i] National Center for Atmospheric Research (NCAR) 4-Channel Chemiluminescence Instrument (Weinheimer et al.,
141 1993)

ᵀ NASA Diode Laser Hygrometer (DLH) (Podolske et al., 2003).

### 2.1.2    AERONET AOD

We use ground-based AOD observations to evaluate our parameterization and simulated AOD. The Aerosol Robotic
Network (AERONET) is a worldwide network that provides long-term sun photometer measured AOD, and is
conventionally considered as the ground truth for evaluating model-simulated (Zhai et al., 2021; Meng et al., 2021;
Jin et al., 2023) or satellite-retrieved AOD (Levy et al., 2013; Wang et al., 2014a; Kahn et al., 2005; Lyapustin et al.,
2018). We use the Version 3 Level 2 database, which includes an improved cloud screening algorithm (Giles et al.,
2019). AOD at 550 nm wavelength, interpolated based on the local Ångström exponent at 440 and 670 nm channels,
is used in this study. For each site, we use data for the year 2017, excluding months with less than 20 days of
measurements and excluding sites with less than 4 months of observations.

### 2.2    GEOS-Chem simulation

We interpret the aircraft observations with the GEOS-Chem chemical transport model (www.geos-chem.org, last
access: 30 October 2022). GEOS-Chem is driven by offline meteorological data from the Goddard Earth Observing
System (GEOS) of the NASA Global Modeling and Assimilation Office (Schubert et al., 1993). We use the high-
performance implementation of GEOS-Chem (GCHP) (Eastham et al., 2018; Bindle et al., 2021)  to examine the
effect of variation in aerosol size on AOD. We also use the TOMAS microphysical scheme, coupled with the standard
GEOS-Chem implementation (GEOS-Chem Classic), to explicitly resolve aerosol microphysics. The bulk and the
microphysics simulations share common emissions and chemical mechanisms. They are both conducted for the year
2017 and driven by MERRA-2 meteorological fields.
The GEOS-Chem aerosol simulation includes the sulfate-nitrate-ammonium system (Fountoukis and Nenes, 2007;
Park, 2004), primary and secondary carbonaceous aerosols (Park et al., 2003; Wang et al., 2014b; Marais et al., 2016;
Pye et al., 2010), sea salt (Jaeglé et al., 2011), and mineral dust (Fairlie et al., 2007). The primary emission data are
from the Community Emissions Data System (CEDS$_{GBD-MAPS}$; McDuffie et al., 2020). Dust emission inventories
include updated natural dust emission (Meng et al., 2021b), and anthropogenic fugitive, combustion, and industrial
dust (AFCID; Philip et al., 2017). Resolution-dependent soil $NO_x$, sea salt, biogenic VOC, and natural dust emissions
are calculated offline at native meteorological resolution to produce consistent emissions across resolution (Weng et
al., 2020; Meng et al., 2021b). Biomass burning emissions use the Global Fire Emissions Database, version 4 (GFED4)
(Van Der Werf et al., 2017). We estimate organic matter (OM) from primary organic carbon using the same OM/OC
parameterizations as applied for DISCOVER-AQ (Philip et al., 2014; Canagaratna et al., 2015). Dry and wet
deposition follows Amos *et al.* (2012), with a standard resistance-in-series dry deposition scheme (Wang et al., 1998).
Wet deposition includes scavenging processes from convection and large-scale precipitation (Liu et al., 2001).
Global relative humidity dependent aerosol optical properties are based on the Global Aerosol Data Set (GADS)
(Kopke P., 1997; Martin et al., 2003) with updates for SNA and OM (Latimer and Martin, 2019), mineral dust (Zhang
et al., 2013), and absorbing brown carbon (Hammer et al., 2016). In the current GEOS-Chem model, the SNA and
OM $R_{eff}$ of particular interest here are based on co-located measurements of aerosol scatter and mass from the
IMPROVE network at U.S. national parks over the period 2000-2010, together with a κ-Kohler framework for aerosol
hygroscopicity (Kreidenweis et al., 2008) as implemented by Latimer and Martin (2019). Aerosol extinction is
calculated as the sum of extinction from each aerosol component with aerosol optical properties listed in Table A1, as
described in Appendix A2.
A global GCHP simulation (Eastham et al., 2018) version 13.0.0 (DOI: 10.5281/zenodo.4618180) that includes
advances in performance and usability (Martin et al., 2022), is conducted on a C90 cubed-sphere grid corresponding
to a horizontal resolution of about 100 km, with a spin-up time of 1 month.
The TOMAS microphysics scheme, coupled with the GEOS-Chem simulation, conserves aerosol mass, and tracks
particles with diameters from approximately 1 nm to 10 μm (Adams and Seinfeld, 2002). Microphysical processes in
TOMAS include nucleation, condensation, evaporation, coagulation, and wet and dry deposition (Adams and Seinfeld,
2002). Nucleation in TOMAS follows a ternary scheme (sulfuric acid, ammonia, and water) when ammonia mixing
ratios are greater than 0.1 ppt; otherwise, a binary nucleation scheme is used (Napari et al., 2002). The nucleation rate
is scaled by $10^{-5}$ to better match the observations (Westervelt et al., 2013). The condensation and evaporation algorithm
is based on a study from Tzivion *et al.* (1989), including interaction with secondary organic aerosol (D'Andrea et al.,
2013). Interstitial coagulation in clouds is also included (Pierce et al., 2015).
For each size bin, TOMAS tracks the mass and number of sulfate, sea salt, black carbon, OC, dust, and water. Primary
sulfate emissions have 2 lognormal modes: 15% of the mass with a number median diameter (NMD) of 10 nm and
geometric standard deviation (σ) of 1.6 and the remainder with a NMD of 70 nm and σ of 2 (Adams and Seinfeld,
2003). The size of emitted carbonaceous particles varies depending on the source: those produced by fossil fuel have
a NMD of 30 nm and σ of 2, while biofuel and biomass burning particles are emitted with a NMD of 100 nm and σ
of 2 (Pierce et al., 2007). Meteorology and most of the emissions in GEOS-Chem-TOMAS follow the bulk simulation,
except that online schemes are used for dust (Zender et al., 2003) and sea salt (Jaeglé *et al.* 2011).
The GEOS-Chem-TOMAS (version 13.2.1. DOI: 10.5281/zenodo.5500717) is used to provide insights into global
scale aerosol size variation and the driving factors. For computational feasibility, a one-year global simulation is
conducted with a horizontal resolution of 4º × 5º and 47 vertical layers from surface to 0.01 hPa. The spin-up time is
1 month. Aerosols are tracked in 15 size bins with particle diameters ranging from about 3 nm to 10 μm. We also
conducted a 2º × 2.5º simulation for October to evaluate the sensitivity of our conclusions to the resolution of the
aerosol microphysics simulation.
**3    Development of a Parameterization of Aerosol Size**
We first examine the aircraft measurements for insight into the observed variation in aerosol size. Then we apply the
size-resolved GEOS-Chem-TOMAS model to extend our analysis to the global scale and identify driving factors of
aerosol size. We subsequently develop and test a parameterization of aerosol size for use in bulk models.

     **3.1     Observed variation in aerosol size**

Figure 1 shows the daily-mean dry effective radius from DISCOVER-AQ and KORUS-AQ as a function of aerosol
mass. Aerosol size, in terms of dry $R_{eff}$, ranges from 90 nm to 179 nm for DISCOVER-AQ, which is generally smaller
than for KORUS-AQ that ranges from 135 nm to 174 nm. $M_{SNAOM}$ from DISCOVER-AQ (1.4 μg/m$^3$ to 27.4 μg/m$^3$)
is also generally less than that from KORUS-AQ (5.5 μg/m$^3$ to 33.2μg/m$^3$). A strong correlation (r = 0.83) between
aerosol size and $M_{SNAOM}$ is evident. $R_{eff}$ from KORUS-AQ is less sensitive to $M_{SNAOM}$ (slope = 1.23) compared to
DISCOVER-AQ (slope = 3.57). The relatively large particle size at low mass concentration during KORUS-AQ might
reflect the influence of aged aerosol transported from upwind (Jordan et al., 2020; Zhai et al., 2021; Nault et al., 2018).

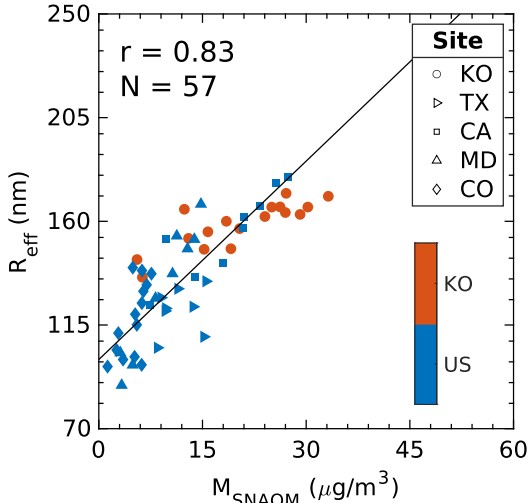


**Figure 1. Airborne measurements of dry effective radius ($R_{eff}$) versus the sum of SNA and OM mass**
**($M_{SNAOM}$) for DISCOVER-AQ (Maryland is abbreviated as MD, California as CA, Texas as TX, Colorado as**
**CO) and for KORUS-AQ (KO) campaigns. Each point represents a daily average for the entire flight profile.**
**Only data with $M_{SNAOM}$ > 4 × Dust mass is used.**

The positive relationship between dry aerosol size and mass of SNA and OM reflects the roles of emission,
condensation, and coagulation in simultaneously increasing aerosol size and mass. This general tendency is also
observed by many other studies (e.g., Sakamoto et al., 2016; Rodríguez et al., 2007; Sun et al., 2012; Bahreini et al.,
2003) despite variable aerosol sources and growth mechanisms. In cities, the joint increases in aerosol size and mass
are usually attributable to anthropogenic emissions and condensation (Tian et al., 2019; Sun et al., 2011; Huang et al.,
2013). In remote areas, biomass burning shifts the particle size distribution toward larger radii due to high emission
rates and coagulation in plumes (Rissler et al., 2006; Ramnarine et al., 2019) that, for example, increase both aerosol
size and mass from the wet season to the dry season in Amazonia (Rissler et al., 2006; Andreae et al., 2015). The
positive relationship between aerosol size and mass suggests the possibility of using aerosol mass as a predictor of
$R_{eff}$.
We examine the ability of the GEOS-Chem bulk model to reproduce the observed extinction. The top panel of Figure
2 compares the measured aerosol extinction profiles to calculated aerosol extinction profiles using default $R_{eff}$. Details
about the calculation are described in Appendix A2. Both measured and calculated extinction profiles exhibit
increasing extinction toward the surface associated with increasing aerosol mass concentrations. However, biases are
apparent for both DISCOVER-AQ and KORUS-AQ. The $R_{eff}$ from KORUS-AQ shown in Figure 1 have a mean value
of 164 nm, larger than the value of 101 nm inferred by Latimer & Martin (2019) based on measurements of aerosol
scatter and mass by the U.S. IMPROVE network. This bias was previously noted by Zhai *et al.* (2021). The mean $R_{eff}$
from DISCOVER-AQ of 138 nm is also larger than the inferred value. This likely reflects representativeness
differences since the DISCOVER-AQ deployments focused on major urban areas during months of high aerosol
loadings, while the IMPROVE measurements were at national parks throughout the year. The middle panel shows the
calculated extinction using the measured aerosol size distribution. Applying the measured aerosol size distribution
addresses most discrepancies between the calculated and measured aerosol extinction profile for both KORUS-AQ
and DISCOVER-AQ. The corresponding discrepancies in AOD estimation also significantly decreased (from 0.09 to
0.03 for DISCOVER-AQ and from 0.17 to 0.02 for KORUS-AQ). The reduced discrepancies support the conclusions
from Zhai *et al.* (2021) that the GEOS-Chem aerosol size is underestimated for KORUS-AQ and motivate
parameterization of $R_{eff}$ for efficient representation of aerosol size for global scale aerosol modeling.

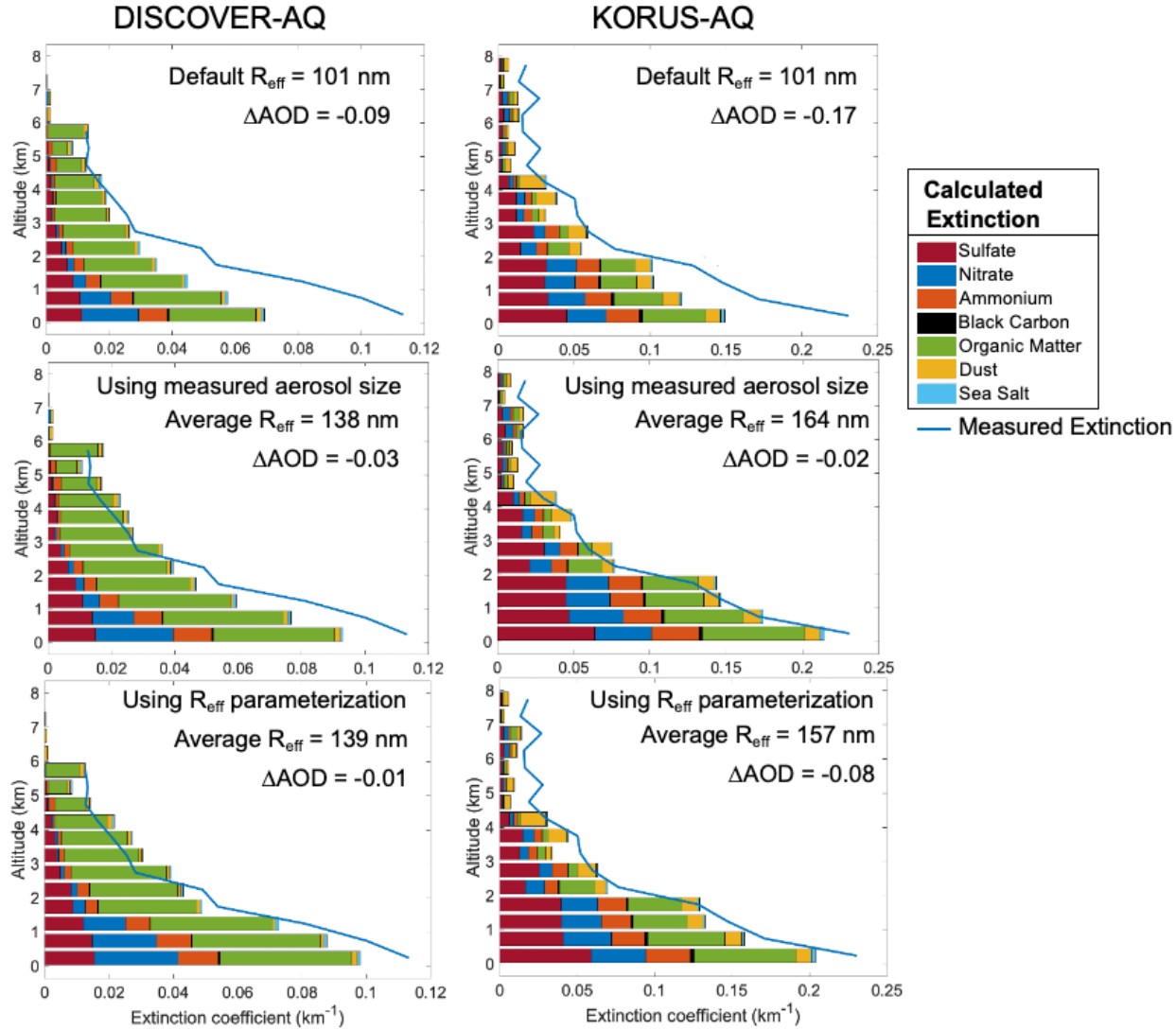

**Figure 2. Aerosol extinction profile for the DISCOVER-AQ and KORUS-AQ aircraft campaigns. Blue lines are the measured extinction profiles. Horizontal bars are calculated extinction using (top) default GEOS-Chem R$_{eff}$, (middle) measured R$_{eff}$, and (bottom) parameterized R$_{eff}$ (described in Section 3.3), together with measured aerosol composition and RH. The aerosol extinction calculation is described in Appendix A.**

### 3.2 Driving factors

Given the strong positive correlation of aerosol mass with aerosol size, we further examine this relationship globally using GEOS-Chem coupled with the TOMAS aerosol microphysics scheme. To focus on areas that are dominated by SNA and OM, we only include grid boxes with M$_{SNAOM}$ > 90% of the aerosol mass. Inspection of the GEOS-Chem-TOMAS size distribution across continental regimes reveals a general tendency for the distribution to shift toward smaller sizes as R$_{eff}$ decreases and toward larger sizes as R$_{eff}$ increases, thus supporting the use of the single summary statistic of R$_{eff}$ for aerosol size. The top panel of Figure 3 shows the geographic distribution of annual mean surface layer dry R$_{eff}$ for grid boxes that meet the criterion. Among the areas of interest, biomass burning regions of Central

Africa, South America, and boreal forests of North America exhibit the highest surface $R_{eff}$ of about 180 nm. Industrial areas such as East Asia and South Asia also exhibit high $R_{eff}$ of about 130 nm, given an abundance of particle emissions and gaseous precursors. The lowest surface $R_{eff}$ of about 80 nm is found in North America, where aerosol mass concentrations are low.

The middle panel of Figure 3 shows the simulated $M_{SNAOM}$ from GEOS-Chem-TOMAS. Enhanced $M_{SNAOM}$ concentrations of over 40 $\mu g/m^3$ are apparent over East Asia and South Asia, reflecting intense anthropogenic emissions. Another $M_{SNAOM}$ hotspot can be seen in Central Africa, driven by biomass burning during the dry season (Van Der Werf et al., 2017; McDuffie et al., 2021) and sometimes exacerbated by anthropogenic emissions (Ngo et al., 2019). Moving from North America to Europe, and then to Asia (defined by boxes in the middle panel), $M_{SNAOM}$ concentrations exhibit a generally increasing tendency (mean value of 11, 17, and 25 $\mu g/m^3$, respectively), consistent with the $R_{eff}$ tendency (mean value of 124, 133, and 136 nm, respectively) in the top panel and aligning with the relationship between aircraft measurements over the U.S. and South Korea.

However, in South America, where $R_{eff}$ is among the highest, $M_{SNAOM}$ is relatively low. This discrepancy motivates the search for other factors, such as aerosol composition, that are associated with aerosol size. In South America, aerosol mass is mostly from natural sources, particularly biomass burning during the dry seasons. $R_g$ for a particle population from biomass burning ranges from 60 nm to 170 nm (Rissler et al., 2006; Reid et al., 2005; Janhall et al., 2010), usually larger than that of primary sulfate aerosol (5 to 35 nm) (Whitey, 1978; Plaza et al., 2011). Therefore, the relative abundance of OM in the total $M_{SNAOM}$ can serve as another predictor of $R_{eff}$. The bottom panel of Figure 3 shows the ratio between OM and SNA mass. In addition to the Amazon basin, the biomass burning regions of Central Africa and boreal forests in Asia and North America are all areas with high OM mass fractions, which contribute to their high $R_{eff}$.

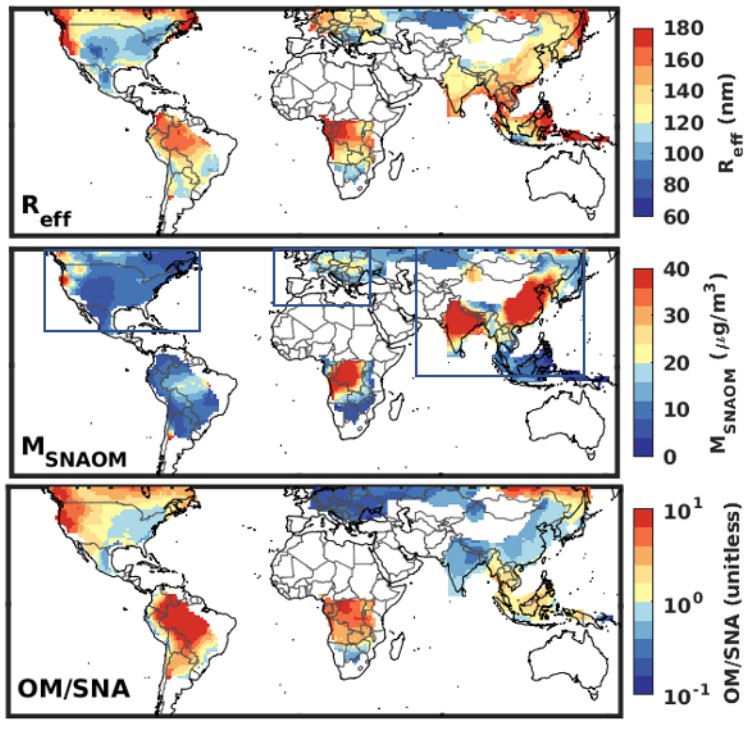

282

**Figure 3: Geographic distribution of GEOS-Chem-TOMAS-simulated annual mean surface layer aerosol properties; (top) $R_{eff}$ when $M_{SNAOM}$ > 90% of aerosol mass, (middle) the sum of SNA and OM mass ($M_{SNAOM}$), and (bottom) OM/SNA.**

### 3.3 Parameterization and evaluation

We use Multiple Linear Regression (MLR) to derive a parameterization of dry $R_{eff}$ for SNA and OM as a function of $M_{SNAOM}$ and OM/SNA. We sample the GEOS-Chem-TOMAS simulation for locations dominated by $M_{SNAOM}$ (> 90%). We include all qualified data (8,569 grid boxes) from the planetary boundary layer (PBL) to focus on this region, while randomly sample 0.5% of simulations in the free troposphere (217,772 grid boxes) to allow the influence of remote regions in the training set. The reason for focusing on the PBL is twofold. First, the PBL generally has the highest aerosol loading that largely determines the columnar mass and AOD (Koffi et al., 2016; Zhai et al., 2021; Tian et al., 2019). Second, the PBL is the domain where most model-measurement difference exists (Figure 2, top panel).

Taking the logarithm of $R_{eff}$ and the logarithm of the two predictors facilitates linear relationships for regression, which yields the initial parameterization

$$R_{eff} = 78.3 M_{SNAOM}^{0.20} \left(\frac{OM}{SNA}\right)^{0.065} \qquad \text{Eqn. (4)}$$

where $R_{eff}$ has units of nm, $M_{SNAOM}$ has units of μg/m$^3$, and OM/SNA is unitless. The $R_{eff}$ parameterization is driven primarily by the mass of SNA and OM, modulated by the ratio of OM to SNA. This equation well represents the variation of $R_{eff}$ during the aircraft campaigns with an $R^2$ of 0.74 (Figure B1, top left). The slope below unity (0.90)

likely reflects the effect of coarse model resolution, which dilutes the particle or precursor concentration in turn
reducing condensation and coagulation growth (AboEl-Fetouh et al., 2022; Ramnarine et al., 2019; Sakamoto et al.,
2016). Adjustment to this parameterization to account for these effects and align the slope with the airborne
measurements rather than the model results in a final parameterization of

$$R_{eff} = 87.0 M_{SNAOM}^{0.20} \left(\frac{OM}{SNA}\right)^{0.065}$$  Eqn. (5)

Figure 4 shows the distribution of dry $R_{eff}$ based on GEOS-Chem-TOMAS and Eqn. (5). Circles in Figure 4 show the
mean values of the sampled GEOS-Chem-TOMAS simulated $R_{eff}$ as a function of simulated $M_{SNAOM}$ concentrations,
ranging from 0.02 to 102 μg/m$^3$, and OM/SNA ranging from 0.13 to 55. Simulated $R_{eff}$ extends from 15 nm when both
$M_{SNAOM}$ and OM/SNA are low (0.09 μg/m$^3$ and 1.3, respectively), up to 282 nm when $M_{SNAOM}$ and OM/SNA are high
(about 44 μg/m$^3$ and 14 respectively). The background color indicates our parameterized $R_{eff}$. A high degree of
consistency exists between the parameterized $R_{eff}$ and simulated $R_{eff}$, especially in the free troposphere where large
gradients in $R_{eff}$ exist, with overall for the troposphere an $R^2$ of 0.72, and a slope of 0.81 (Figure B1, bottom right). At
the lower end of $R_{eff}$, the agreement between simulation and the parameterization can also be found in Figure B1,
which shows that the small $R_{eff}$ are reproduced by the parameterization. Despite the overall consistency, a few
differences exist. When aerosol mass concentration is high, the parameterization tends to yield a higher $R_{eff}$ than in
the GEOS-Chem-TOMAS simulation, since the adjustment using aircraft measurements led to 11% increase in $R_{eff}$.
At $M_{SNAOM}$ near 10 μg/m$^3$ and OM/SNA near 10, the simulation indicates higher $R_{eff}$ than the parameterization,
reflecting dilution downwind of biomass burning that reduces the aerosol mass concentration but has less influence
on particle size in GEOS-Chem-TOMAS (Park et al., 2013; Rissler et al., 2006; Sakamoto et al., 2016). A 10-20%
underestimation in the parameterization at low OM/SNA reflects the advection and dilution downwind of urban areas
and in the free troposphere (Yue et al., 2010; Asmi et al., 2011). Evaluation of our parameterization versus the GEOS-
Chem-TOMAS simulation of 2º × 2.5º for October yields similar results but explains an additional 14% of the variance
in simulated $R_{eff}$, providing additional evidence of the fidelity of the parameterization.

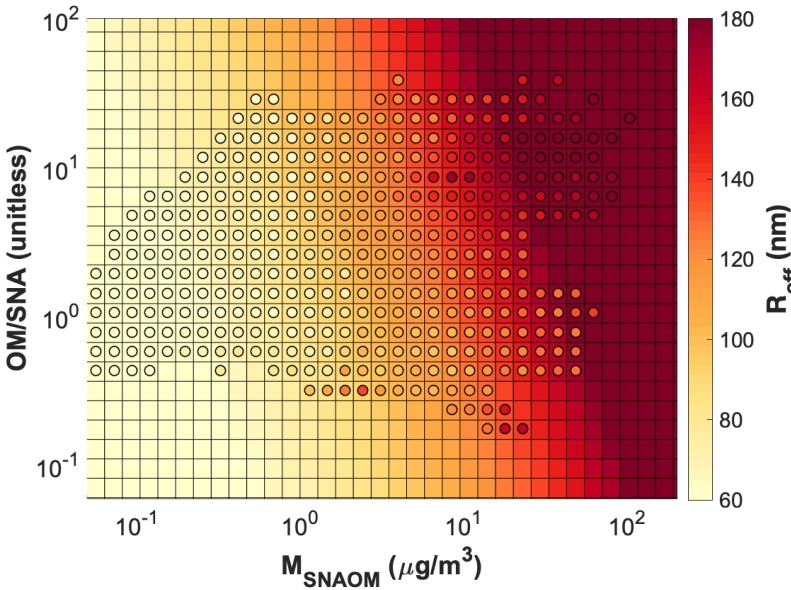

**Figure 4. Dry $R_{eff}$ as a function of $M_{SNAOM}$ and OM/SNA when SNA and OM are dominant (>90%). Each circle represents the mean value of the GEOS-Chem-TOMAS simulated $R_{eff}$ in each bin. Background color indicates the parameterized $R_{eff}$.**

When applied to the airborne measurements, this parameterization only slightly overestimates the measured $R_{eff}$ from DISCOVER-AQ (139 nm vs. 138 nm) and slightly underestimates $R_{eff}$ from KORUS-AQ (157 nm vs. 164 nm). Discrepancies between calculated and measured extinction from aircraft campaigns are largely reduced (Figure 2, bottom panel) with AOD biases of 0.01 and 0.08 for DISCOVER-AQ and KORUS-AQ, respectively. Minor differences are still present in aerosol extinction above 4 km for KORUS-AQ, but a physical explanation remains elusive since the calculated extinction is biased even if measured aerosol size and composition are used; instrument uncertainties may play a role. Nonetheless, effects on columnar AOD from these disagreements aloft are minor (<5%).

We then apply Eqn. (5) to a GEOS-Chem bulk simulation to calculate $R_{eff}$ and AOD. The top panel of Figure 5 shows the annual mean dry $R_{eff}$ for surface SNA and OM aerosol. The parameterized $R_{eff}$ is usually higher than the default value of about 100 nm in GEOS-Chem over land, and lower than that over the ocean, with a normalized root mean square deviation (NRMSD) of 43.8%. The parameterized $R_{eff}$ is the highest in biomass burning regions in South America and Central Africa, as well as industrial regions in Asia, similar to the pattern found in the GEOS-Chem-TOMAS simulation. The parameterized $R_{eff}$ and its horizontal variation diminish with altitude (Figure B2), with the mean $R_{eff}$ of 85 nm at the surface decreasing by 18.8% to 69 nm at about 5 km. By design, the parameterization has little effect in regions and seasons where and when $M_{SNAOM}$ is not dominant, since the parameterization only affects $R_{eff}$ of SNA and OM.

The middle panel of Figure 5 shows the simulated AOD, with the corresponding difference between the base simulation and the updated simulation in the bottom panel. To accommodate the parameterized $R_{eff}$, a look-up table with a wide range of $R_{eff}$ (0.02 μm to 1.7 μm) and the corresponding extinction efficiencies for OM and SNA is created

based on Mie Theory (Mishchenko et al., 2002, 1999). This update generally increases aerosol mass scattering by
increasing the mass extinction efficiency, in turn, increasing AOD over regions with strong anthropogenic sources,
such as East Asia (by 0.10, 28.3%) and South Asia (by 0.14, 31.1%). It also slightly increases AOD over regions
influenced by wildfires, such as South America (by 0.02, 19.7%), Central Africa (by 0.03, 22.7%), and the boreal
forests in Europe (by 0.01, 9.9%). Most increases occur near the surface (Figure B3), where the highest aerosol mass
loading and mass extinction efficiency exist. The NRMSD between original and updated GEOS-Chem simulated AOD
is 18.9% globally, and 25.6% over continents.

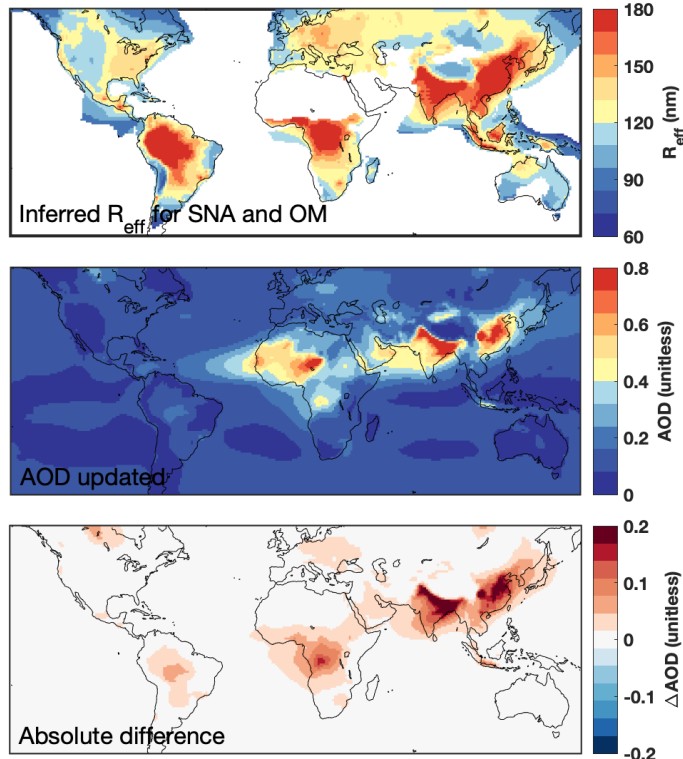


**352 Figure 5. (Top) Surface dry $R_{eff}$ for SNA and OM calculated using Eqn. (5) and GEOS-Chem bulk model**

**353 simulated SNA and OM mass. $R_{eff}$ is shown when $M_{SNAOM}$ is greater than 80% of the total aerosol mass.**

**354 (Middle) The GEOS-Chem simulated AOD using inferred $R_{eff}$. (Bottom) the absolute difference between**

**355 updated AOD and default AOD using dry $R_{eff}$ = 101 nm.**

Although $R_{eff}$ is only one of many processes affecting AOD, we evaluate the effect of the parameterization on the
GEOS-Chem simulation of AOD to assess its implications. The left and middle panels of Figure 6 show the
discrepancy between GEOS-Chem simulated AOD and AERONET AOD as a function of the parameterized surface
$R_{eff}$ for SNA and OM. The simulation using the default $R_{eff}$ (Figure 6, left panel) slightly overestimates AOD at sites
with small parameterized $R_{eff}$ and underestimates AOD at sites with large parameterized $R_{eff}$. The overestimates occur
primarily in western Europe where SNA and OM concentrations are low, while the underestimates happen mainly
over industrial regions in East Asia, Southeast Asia, and biomass burning areas in South America and Central Africa,
where the SNA and OM mass loading are high (Figure B4). The underestimates are mitigated when applying the
parameterized $R_{eff}$ in GEOS-Chem (Figure 6, middle panel), yielding increased consistency between the measured
(AERONET) AOD and simulated AOD (Figure 6, right; $R^2$ change from 0.68 to 0.74, slope from 0.75 to 0.94).

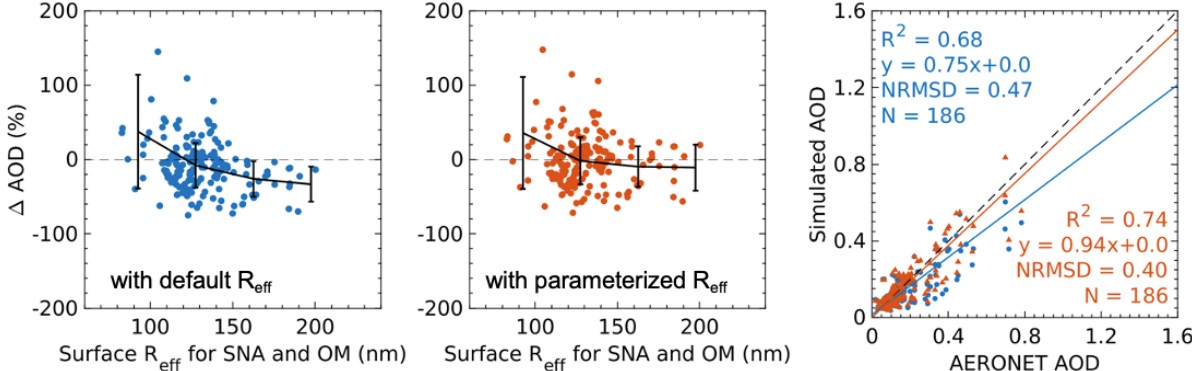


**Figure 6. (Left and middle) Percent increase in GEOS-Chem simulated AOD minus AERONET AOD as a**
**function of parameterized surface dry $R_{eff}$ for SNA and OM. Black lines represent the mean values of ΔAOD**
**in each 35 nm bin; error bars represent the corresponding standard deviation. (Right) Scatter plot of**
**AERONET versus simulated AOD with the default $R_{eff}$ (blue dots, line, and text), and with the parameterized**
**$R_{eff}$ (red dots, line, and text). The 1:1 line is dashed. NRMSD is the normalized root mean square deviation**
**between the two datasets. N is the number of points in each dataset.**
**4    Conclusion**
Aerosol size strongly determines mass scattering efficiency with implications for calculation of aerosol optical
properties. Prior work found that the global mean dry aerosol size used in a bulk aerosol model induced low bias
versus measured extinction in a region with a high aerosol loading (Zhai et al., 2021). We interpreted aircraft
measurements from DISCOVER-AQ and KORUS-AQ with a chemical transport model (GEOS-Chem) to better
understand regional variation in aerosol size. The measurements had a strong positive correlation (r = 0.83) between
aerosol size and mass of sulfate-nitrate-ammonium (SNA) and organic matter (OM), reflecting the high condensation
and coagulation rates where emissions of particles and the gaseous precursors are abundant, indicating the possibility
of using aerosol mass as a predictor of aerosol size.
To gain a broader perspective of the global variation in aerosol size, we used the TOMAS microphysics package of
the GEOS-Chem model to simulate the monthly mean aerosol mass, composition, and size distribution. We used
effective radius ($R_{eff}$) as a surrogate of aerosol size and examined its relationship with aerosol mass and components
where SNA and OM were dominant. We found that the sum of SNA and OM concentration, and the ratio between
them, were the major predictors of $R_{eff}$. We used GEOS-Chem-TOMAS model output to derive a parameterization of
$R_{eff}$, which well reproduced $R_{eff}$ measured from the aircraft campaigns ($R^2$ = 0.74). When applied in the bulk GEOS-
Chem high-performance model, the parameterization tended to increase $R_{eff}$ of SNA and OM over regions with high
concentrations of SNA and OM, and decrease $R_{eff}$ elsewhere relative to the standard model. This led to a global
normalized root mean square deviation (NRMSD) of 43.8% between the original and updated surface $R_{eff}$. The
parameterized $R_{eff}$ tended to increase the vertical gradient in extinction relative to the standard model, due to the
decrease in $R_{eff}$ with altitude. The NRMSD of global mean AOD between the original and updated simulations was
18.9%, with the most significant regional AOD increase of 0.14 in South Asia, where aerosol mass loadings are high.
This parameterization led to improved consistency of GEOS-Chem simulated AOD with AERONET AOD ($R^2$ from
0.68 to 0.74; slope from 0.75 to 0.94), by increasing AOD at high $R_{eff}$.
Overall, the simple parameterization of $R_{eff}$ derived in this study improved the accuracy in modeling aerosol optical
properties without imposing additional computational expense. Other chemical transport models and modeling of
other size-related processes, such as heterogeneous chemistry, photolysis frequencies, and dry deposition, may also
benefit from the parameterized $R_{eff}$. Future work could include additional parameters to better summarize the aerosol
size distribution. Further developments in computational efficiency of aerosol microphysics models and more
abundant measurements of aerosol size and optical properties would both offer opportunities for further advances.

*Data availability.* AERONET data can be found at https://aeronet.gsfc.nasa.gov/. Aircraft data during DISCOVER-AQ are available at https://asdc.larc.nasa.gov/project/DISCOVER-AQ. KORUS-AQ data can be found at https://doi.org/10.5067/Suborbital/KORUSAQ/DATA01.

*Author contributions.* HZ and RVM designed the study. HZ performed the data analysis and model simulations with contributions from BC, SZ, CL, LB, JRP, IS, DC, and RYWC. BEA, LDZ, JWH, RAF, CAH, JLJ, PCJ, JED, JSS, AW, and BAN contributed to KORUS-AQ and DISCOVER-AQ campaign measurements. HZ and RVM wrote the paper with input from all authors.

*Competing interests.* The contact author has declared that neither they nor their co-authors have any competing interests.

*Financial support.* This work was supported by NASA Grant 80NSSC21K1343. JRP was funded by the US NSF Atmospheric Chemistry program, under grant AGS-1950327. JLJ and PCJ were supported by NASA Grant 80NSSC21K1451 and NNX15AT96G. BAN was supported by NASA Grant 80NSSC22K0283. JED was supported by NASA Grant NNX15AT88G.

## Appendix A

### A1 Application of spatially and temporally varying OM/OC ratio

The top panel of Figure A1 shows scatter plots of the estimated and measured OM/OC and OM during the KORUS-AQ campaign. The estimation is obtained by applying to OC measurement a $NO_2$ inferred OM/OC from (Philip et al., 2014), with a subsequent correction factor of 1.09 suggested by Canagaratna et al. (2015). Estimated OM is compared with measured OM by AMS during the campaign. Overall consistency is evident between $NO_2$-derived OM/OC and measured OM/OC. The agreement is better below 500 m than above (left panel, $R^2 = 0.62$ vs. 0.33). The discrepancy at high altitudes is mainly due to the low $NO_2$ (<0.2 ppbv), where the Philip *et al.* (2014) equation is not applicable. An average OM/OC ratio (2.1) is applied in this case. A high degree of consistency exists between the estimated OM and measured OM, with $R^2 = 0.99$ and slope = 0.91 for data from all altitudes (right panel), thus supporting the use of estimated OM in our analyses. The bottom left panel compares the vertical profile of the estimates and measurements, yielding overall consistency.

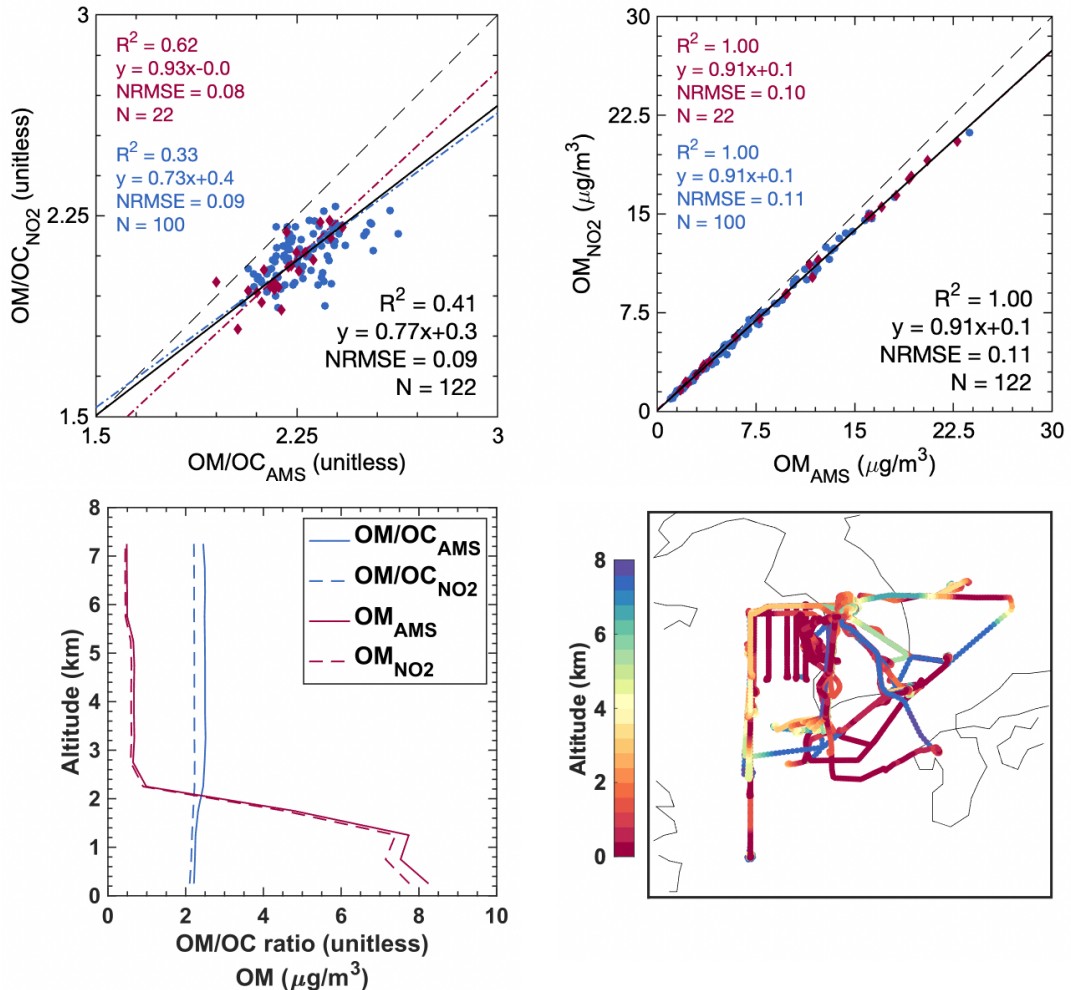


**Figure A1. Scatter plots of estimated and measured OM/OC (top left) and OM (top right) during KORUS-**
**AQ. Each point represents a mean value of AMS measurement for a 1-hour interval. Red diamonds, lines,**
**and texts represent data from 0-500 m altitude. Blue dots, lines, and text represent data above 500 m from the**
**ground. Black solid lines and texts represent the line of best fit for all the data. The 1:1 line is dashed.**
**NRMSD is the normalized root mean square deviation between the two datasets. N is the number of points in**
**each dataset. (Bottom left) Mean values of OM/OC and OM from measurements and estimations along the**
**altitude. (Bottom right) Flight tracks during KORUS-AQ.**

**A2 Aerosol Extinction Calculation in GEOS-Chem**

Extinction (Ext) of radiation by aerosols is represented as the sum of extinction due to each of the aerosol components using the following equation:

$$Ext_k = \frac{3Q_{ext,k}M_k}{4\rho_k R_{eff,k}}$$

Eqn. (3)

where subscript k indicates the property for the $k^{th}$ component. $R_{eff}$ is the effective radius defined as the area weighted mean radius. $Q_{ext}$ is the area-weighted mean extinction efficiency. M is the aerosol mass loading per unit volume. $\rho$ is the aerosol density. Aerosol optical depth (AOD) is the integral of aerosol extinction across the vertical domain.

For each component, extinction is calculated for assumed log-normal size distribution with corresponding dry geometric mean radius $R_g$ and geometric standard deviation $\sigma$, hygroscopicity, refractive index (RI), and density ($\rho$) for individual aerosol components, as listed in Table A1. Sulfate, nitrate, and ammonium are grouped into SNA for convenience. $R_{eff}$ and $Q_{ext}$ are calculated using Mie Theory (Mishchenko et al., 1999, 2002) based on assumptions in aerosol size and RI. Hygroscopicity for SNA and OM is represented using a κ-Kohler hygroscopic growth scheme (Kreidenweis et al., 2008) as implemented by (Latimer and Martin, 2019).

**Table A1. Dry aerosol properties in GEOS-Chem bulk model**

| Aerosol components | $R_g$, μm | σ | Hygroscopicity | Refractive Index (dry, 550 nm) | ρ, g cm$^{-3}$ | $R_{eff}$, μm | $Q_{ext}$ |
|---|---|---|---|---|---|---|---|
| SNA | 0.058 | 1.6 | $\kappa = 0.61$ | $1.53 – 6.0\times10^{-3}i$ | 1.7 | 0.101 | 0.603 |
| OM | 0.058 | 1.6 | $\kappa = 0.1$ | $1.53 – 6.0\times10^{-3}i$ | 1.3 | 0.101 | 0.603 |

     **Appendix B**

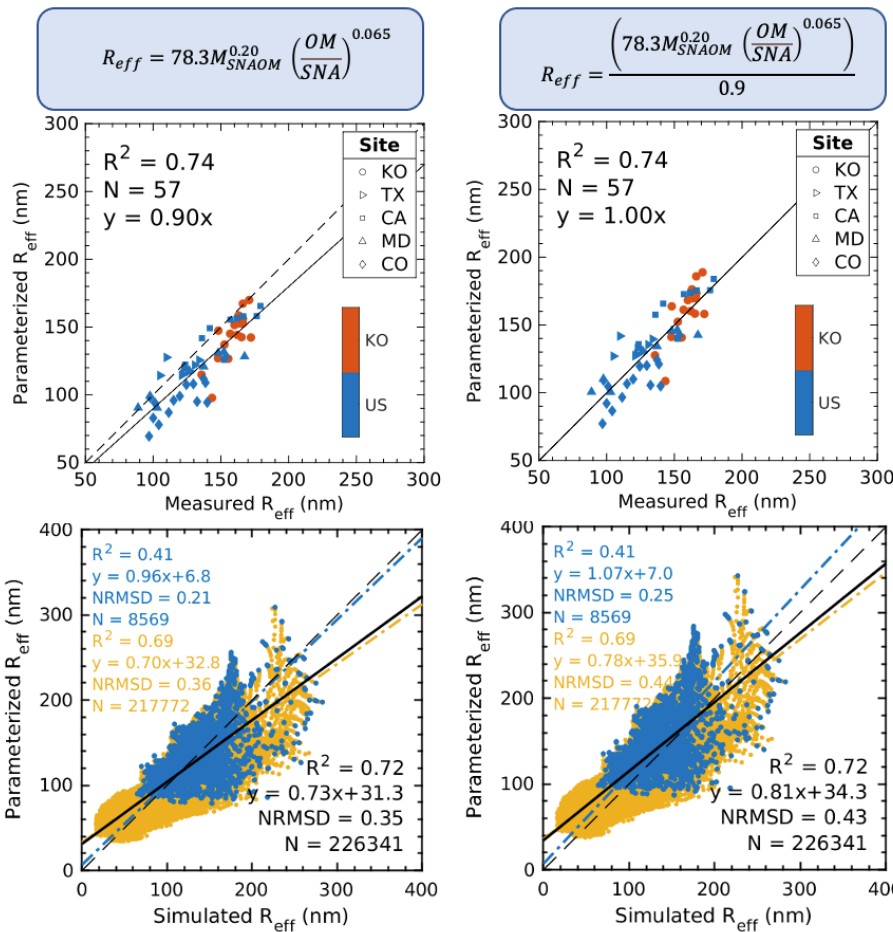

452

**Figure B1. (Top) Scatter plot of parameterized R_eff and measured R_eff from DISCOVER-AQ and KORUS-**
**AQ. Each point represents a daily mean measurement. (Bottom) Scatter plot of parameterized R_eff and**
**GEOS-Chem-TOMAS simulated R_eff for the planetary boundary layer (blue dots, line, and texts), and for the**
**free troposphere (yellow dots, line, and texts). Black solid lines and the texts indicate the entire troposphere**
**with the sum of SNA and OM > 90% of aerosol mass. The 1:1 line is dashed. NRMSD is the normalized root**
**mean square deviation between the two datasets. N is the number of points in each dataset. The left panel**
**indicates the original parameterization from multiple linear regression. The right panel shows the adjusted**
**parameterization using aircraft measurements.**

461

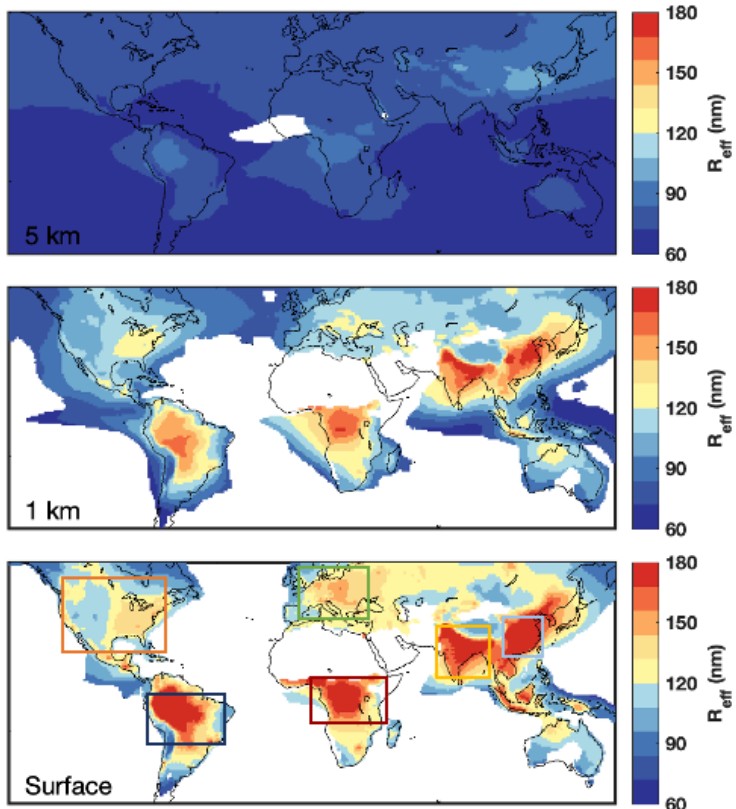

462

**Figure B2. Annual mean $R_{eff}$ for SNA and OM at (top) about 5 km, (middle) about 1 km, and (bottom) surface, calculated using Eqn. (5) and simulated SNA and OM mass from GEOS-Chem bulk model. $R_{eff}$ is shown only if $M_{SNAOM}$ is greater than 80% of the total aerosol mass. Boxes in the bottom panel define regions referred to in Figure B3.**

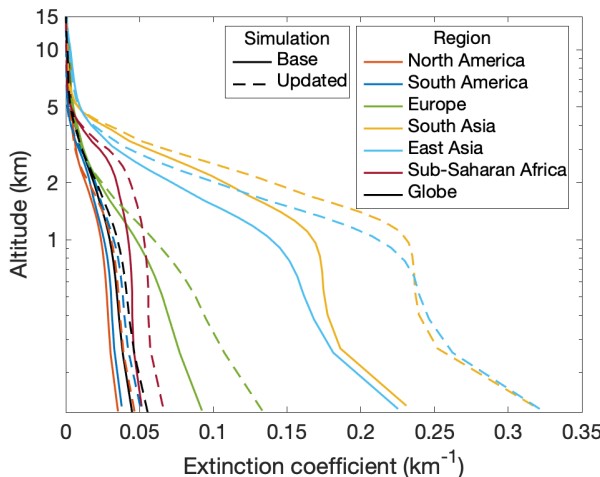


**Figure B3. Global and regional aerosol extinction coefficient simulated by GEOS-Chem bulk model with original $R_{eff}$ (solid lines) and parameterized $R_{eff}$ (dashed lines). Regions are defined by the boxes in Figure B2.**


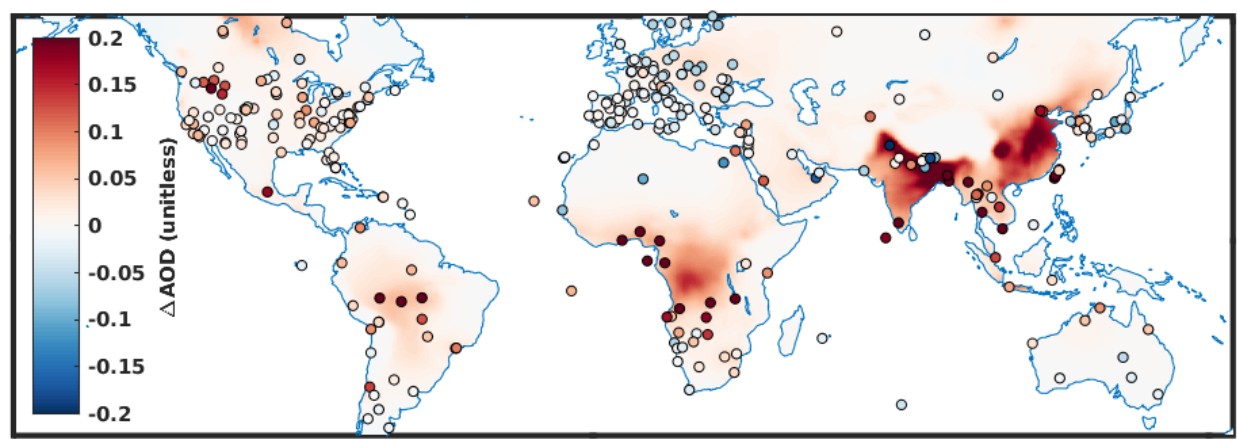

**Figure B4. Difference between AERONET AOD minus default GEOS-Chem simulated AOD (dots) and**
**difference between simulated AOD with the parameterized $R_{eff}$ minus AOD with default $R_{eff}$ (background).**

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
