# Peer review of "Parameterization of Size of Organic and Secondary Inorganic Aerosol for"

_EGUsphere, 2022_

## Referee Comment (RC1)

Aerosol particle size has important influence on its optical properties as well as radiative and climatic effects. To develop a more computationally efficient aerosol particle size parameterization in a global model is of great importance for modelling and remote sensing. The development of the $R_{eff}$ parameterization for OM and SNA combining observations and GEOS-Chem-TOMAS simulation in this study is of good innovative and utility, though there are some inadequacies in the development of the parameterization as well as in its validation. I recommend it can be accepted after the following major revisions.

Major issues:

1. The main demonstration focuses on the time and area dominated by SNA and OM and does not provide sufficient proof of the simulation ability of the new parameterization. In addition, the parameterization is still very deficient for the extinction simulations at upper levels and low aerosol concentrations (as shown on the right side of Fig. 2, many upper level simulated extinction is only about 1/3 of the measured)

2. Some of the statements have inconsistencies between the figures and the text (some of which are pointed out in minor comments), and it is advisable to provide visual illustrations or relevant references when explaining the physics behind the issues in the pictures (e.g., Lines 298-302).

Minor comments:
1. Lines 41-42. Please explain the "strong size dependence" of the number of CCN.

2. Lines 61-62. Only 3 of the 10 models contain online size-resolved aerosol microphysics does not visually illustrate its computational cost, and it may be more convincing to provide comparative results of computational speed or other references.

3. Lines 85-86. Provide references to the dominance of OM and SNA in fine aerosol composition in populated areas.

4. Line 206. Compared to DISCOVER-AQ, $M_{SNAOM}$ in KORUS-AQ results is not significantly sensitive to aerosol size, better to quantify the correlation and provide an explanation.

5. Top of Figure 3. The use of such a low differentiation color for two types of data in a graph makes it difficult for the reader to access the information, and it might be better to use a different marker or a more differentiated color.

6. Line 257. The tendency of $M_{SNAOM}$ and $R_{eff}$ do not look consistent from the graph, and it would be better to provide some quantitative illustration

7. Lines 265-267. The demonstration of OM mass fractions as a driving factor is not rigorous enough, for example, in the area of high OM mass fractions in Central Africa, $M_{SNAOM}$ and Reff have similar distribution in general rather than "high $R_{eff}$ despite the low $M_{SNAOM}$" as stated in the paper.

8.  Line 273. Is it $R_{eff}$ for OM and SNA or for all aerosols?

9.  Top of Figure 5. Same as Figure 3

10. Lines 315-316. $R_{eff}$ magnitude and distribution are not as "well represented" as stated in the paper. The bulk underestimates the hotspot values of TOMAS and their distributions in North America and Europe show significant differences (top of Figure 3 and Figure 5), which need to be further elucidated or improved to eliminate this difference.

11. Lines 316-317. $R_{eff}$ decreases with height, and it is not reasonable to use the $R_{eff}$ value to illustrate the diminished difference at different $R_{eff}$ levels. It is more appropriate to use parameters such as percentage change to express it.

12. Lines 318-327. It is better to combine the satellite-retrieved AOD rather than only using simulated data to illustrate the changes in the updated compared to the base.

13. Lines 335-336. Who does "as a function of the parameterized surface $R_{eff}$ for SNA and OM" refer to specifically?

14. Figure B4. The figure only shows the difference between AERONET and the default GEOS-Chem simulation and the difference between the new parameterization and the default simulation, which does not capture the impact of the new parameterization, so why not directly show the difference between AERONET or other observations with the new parameterization and the default simulation?

---

## Author Comment (AC1)

We thank the referees for their insightful comments. We address each one below and improve the manuscript accordingly. Please find our responses to referees' comments in blue. Newly added references are listed at the end of this document.

**RC1 by Referee #1**

Aerosol particle size has important influence on its optical properties as well as radiative and climatic effects. To develop a more computationally efficient aerosol particle size parameterization in a global model is of great importance for modelling and remote sensing. The development of the $R_{eff}$ parameterization for OM and SNA combining observations and GEOS-Chem-TOMAS simulation in this study is of good innovative and utility, though there are some inadequacies in the development of the parameterization as well as in its validation. I recommend it can be accepted after the following major revisions.

Major issues:

1. The main demonstration focuses on the time and area dominated by SNA and OM and does not provide sufficient proof of the simulation ability of the new parameterization. In addition, the parameterization is still very deficient for the extinction simulations at upper levels and low aerosol concentrations (as shown on the right side of Fig. 2, many upper level simulated extinction is only about 1/3 of the measured)

   To address the concern about simulation ability with the new parameterization, we added discussions in section 3.3. Indeed, we focus on the time and area dominated by SNA and OM to address the manuscript scope as defined by the title and by the abstract in lines 26-27, "investigate the variation in aerosol size when organic matter (OM) and sulfate-nitrate-ammonium (SNA) are the dominant aerosol components." The overall performance for all three independent evaluations is promising as summarized in the abstract for airborne measurements ($R^2 = 0.74$, slope = 1.00), versus a simulation ($R^2 = 0.72$, slope = 0.81), and in line 33-34, "improves the agreement between the simulated aerosol optical depth (AOD) and the ground-measured AOD from the Aerosol Robotic Network (AERONET; $R^2$ from 0.68 to 0.73, slope from 0.75 to 0.96)." The performance of the GEOS-Chem model elsewhere is largely unchanged and beyond the scope of the work. We added to the 4th paragraph in section 3.3, lines 338-340: "By design, the parameterization has little effect in regions and seasons where and when $M_{SNAOM}$ is not dominant, since the parameterization only affects $R_{eff}$ of SNA and OM."

   To address the concern about the upper levels and low aerosol concentrations, we revised the second paragraph in section 3.3, lines 325-331 to: "When applied to the airborne measurements, this parameterization slightly overestimates the measured $R_{eff}$ from DISCOVER-AQ (139 nm vs. 138 nm) and slightly underestimates $R_{eff}$ from KORUS-AQ (157 nm vs. 164 nm). Discrepancies between calculated and measured extinction from aircraft campaigns are largely reduced (**Error! Reference source not found.**, bottom panel) with AOD biases of 0.01 and 0.08 for DISCOVER-AQ and KORUS-AQ, respectively. Minor differences are still present in aerosol extinction above 4 km for KORUS-AQ, but a physical explanation remains elusive since the calculated extinction is biased even if measured aerosol size and composition are used; instrument uncertainties may play a role. Nonetheless, effects on columnar AOD from these disagreements aloft are minor (<5%)."

2. Some of the statements have inconsistencies between the figures and the text (some of which are pointed out in minor comments), and it is advisable to provide visual illustrations or relevant references when explaining the physics behind the issues in the pictures (e.g., Lines 298-302).

To provide visual illustrations, we updated Figure 3 and Figure 5 to more clearly illustrate the trends of the three variables (as shown in responses 5 and 9 below). We calculated the mean $R_{eff}$ and $M_{SNAOM}$ in areas in the boxed regions of Figure 3 and revised lines 269-272 to: "Moving from North America to Europe, and then to Asia (defined by boxes in the middle panel in Figure 3), $M_{SNAOM}$ concentrations exhibit a generally increasing tendency (mean value of 11, 17, and 25 µg/m$^3$, respectively), consistent with the $R_{eff}$ tendency (mean value of 124, 133, and 136 nm, respectively) in the top panel and aligning with the relationship between aircraft measurements over the U.S. and South Korea."

We added additional references and details after line 214 to explain different trends in DISCOVER-AQ data and KORUS-AQ data: "$R_{eff}$ from KORUS-AQ is less sensitive to $M_{SNAOM}$ (slope = 1.23) compared to DISCOVER-AQ (slope = 3.57). The relatively large particle size at low mass concentration during KORUS-AQ might reflect the influence of aerosol transport from upwind (Jordan et al., 2020; Zhai et al., 2021; Nault et al., 2018)."

We also added references to lines 314-318: "At $M_{SNAOM}$ near 10 µg/m$^3$ and OM/SNA near 10, the simulation indicates higher $R_{eff}$ than the parameterization, reflecting dilution downwind of biomass burning that reduces the aerosol mass concentration but has less influence on particle size in GEOS-Chem-TOMAS(Park et al., 2013; Rissler et al., 2006; Sakamoto et al., 2016). A 10-20% underestimation in the parameterization at low OM/SNA reflects the advection and dilution downwind of urban areas and in the free troposphere (Yue et al., 2010; Asmi et al., 2011)."

Minor comments:

1. Lines 41-42. Please explain the "strong size dependence" of the number of CCN.

To explain the strong size dependence of the CCN, we rephrased lines 41-42: "Both direct and indirect aerosol radiative forcing are sensitive to aerosol size, as aerosol size affects the interaction between particles and radiation, and the rate at which a particle grows to a cloud droplet (Adams and Seinfeld, 2002; Faxvog and Roessler, 1978; Mishchenko et al., 2002; Emerson et al., 2020)."

2. Lines 61-62. Only 3 of the 10 models contain online size-resolved aerosol microphysics does not visually illustrate its computational cost, and it may be more convincing to provide comparative results of computational speed or other references.

In response to this comment, we added references at added in lines 61-62: "For example, the wall clock time increases by about 2.5 times when APM is enabled in GEOS-Chem CTM relative to the bulk model (GCST et al., 2023). Only 3 of the 10 models that included aerosols, studied by the Atmospheric Chemistry and Climate Model Intercomparison Project, include online size-resolved aerosol microphysics, reflecting its computational cost and complexity (Lamarque et al., 2013; Liu et al., 2012; Szopa et al., 2013; Kodros and Pierce, 2017)."

3. Lines 85-86. Provide references to the dominance of OM and SNA in fine aerosol composition in populated areas.

We added as references 3 studies on North American and Asian aerosol composition to lines 85-86 (now lines 87-88): "We focus on OM and SNA, which dominate fine aerosol composition in populated areas (Weagle et al., 2018; Geng et al., 2017; Meng et al., 2019; Van Donkelaar et al., 2019; Li et al., 2017)."

4. Line 206. Compared to DISCOVER-AQ, $M_{SNAOM}$ in KORUS-AQ results is not significantly sensitive to aerosol size, better to quantify the correlation and provide an explanation.

We revised the text to quantify and explain as follows in line 214: "$R_{eff}$ from KORUS-AQ is less sensitive to $M_{SNAOM}$ (slope = 1.23) compared to DISCOVER-AQ (slope = 3.57). The relatively large particle size at low mass concentration during KORUS-AQ might reflect the influence of aerosol transport from upwind (Jordan et al., 2020; Zhai et al., 2021; Nault et al., 2018)."

5. Top of Figure 3. The use of such a low differentiation color for two types of data in a graph makes it difficult for the reader to access the information, and it might be better to use a different marker or a more differentiated color.

We updated Figure 3, removing the color intensity dimension and now use a more differentiated color scheme as below:

[Figure]

Figure S-1. Revised Figure 3 in the manuscript: Geographic distribution of GEOS-Chem-TOMAS-simulated annual mean surface layer aerosol properties; (top) $R_{eff}$ when $M_{SNAOM}$ > 90% of aerosol mass, (middle) the sum of SNA and OM mass ($M_{SNAOM}$), and (bottom) OM/SNA.

6. Line 257. The tendency of $M_{SNAOM}$ and $R_{eff}$ do not look consistent from the graph, and it would be better to provide some quantitative illustration.

   To provide quantification, we calculated the mean $R_{eff}$ and $M_{SNAOM}$ in areas squared in Figure 3 and revised lines 269-272 to: "Moving from North America to Europe, and then to Asia (defined by squares in the middle panel in Figure 3), $M_{SNAOM}$ concentrations exhibit a generally increasing tendency (mean value of 11, 17, and 25 µg/m$^3$, respectively), consistent with the $R_{eff}$ tendency (mean value of 124, 133, and 136 nm, respectively) in the top panel and aligning with the relationship between aircraft measurements over the U.S. and South Korea."

7. Lines 265-267. The demonstration of OM mass fractions as a driving factor is not rigorous enough, for example, in the area of high OM mass fractions in Central Africa, $M_{SNAOM}$ and Reff have similar distribution in general rather than "high $R_{eff}$ despite the low $M_{SNAOM}$" as stated in the paper.

   Thank you for catching this. We removed "despite the low $M_{SNAOM}$" from the sentence (line 281).

8. Line 273. Is it $R_{eff}$ for OM and SNA or for all aerosols?

   It is $R_{eff}$ for OM and SNA. To clarify in the text, we added this information to the sentence (line 287): "We use Multiple Linear Regression (MLR) to derive a parameterization of dry $R_{eff}$ for SNA and OM as a function of $M_{SNAOM}$ and OM/SNA" to clarify this.

9. Top of Figure 5. Same as Figure 3

   We revised Figure 5, similar to that for Fig. 3 as discussed in item 5.:

[Figure]

Figure S-2. Revised Figure 5 in the manuscript. (Top) Surface dry $R_{eff}$ for SNA and OM calculated using Eqn. (5) and GEOS-Chem bulk model simulated SNA and OM mass. $R_{eff}$ is shown when $M_{SNAOM}$ is greater than 80% of the total aerosol mass. (Middle) The GEOS-Chem simulated AOD using inferred $R_{eff}$. (Bottom) the absolute difference between updated AOD and default AOD using dry $R_{eff}$ = 101 nm.

10. Lines 315-316. $R_{eff}$ magnitude and distribution are not as "well represented" as stated in the paper. The bulk underestimates the hotspot values of TOMAS and their distributions in North America and Europe show significant differences (top of Figure 3 and Figure 5), which need to be further elucidated or improved to eliminate this difference.

    To further elucidate in the text about these differences, we rephrased lines 335-337 to: "The parameterized $R_{eff}$ is the highest in biomass burning regions in South America and Central Africa, as well as industrial regions in Asia, similar to the pattern found in the GEOS-Chem-TOMAS simulation."

    The discrepancies between the TOMAS simulated $R_{eff}$ and the parameterized $R_{eff}$ using the bulk model are due to inevitable differences between GEOS-Chem-TOMAS and the bulk model: For example, differences in the dust and sea salt emissions since the offline dust and sea salt schemes are not yet available for GEOS-Chem-TOMAS. GEOS-Chem-TOMAS was run at a lower resolution (4×5º vs. 1º). Aerosol microphysics processes in GEOS-Chem-TOMAS also allow size-dependent deposition which affects aerosol concentration. All these resulted in differences in aerosol mass loading and composition between the bulk and the GEOS-Chem-TOMAS model, and therefore the difference between Figure 3 and Figure 5.

11. Lines 316-317. $R_{eff}$ decreases with height, and it is not reasonable to use the $R_{eff}$ value to illustrate the diminished difference at different $R_{eff}$ levels. It is more appropriate to use parameters such as percentage change to express it.

We revised the text to acknowledge and discuss this change with altitude. Lines 337-338 now state: "The parameterized Reff and its horizontal variation diminish with altitude (Figure B2), with the mean $R_{eff}$ of 85 nm at the surface decreasing by 18.8% to 69 nm at about 5 km"

12. Lines 318-327. It is better to combine the satellite-retrieved AOD rather than only using simulated data to illustrate the changes in the updated compared to the base.

In response to this concern, we added this clarification in section 2.1.2, lines 144-148: "We use ground-based AOD observations to evaluate our parameterization and simulated AOD. Aerosol Robotic Network (AERONET) is a worldwide network that provides long-term sun photometer measured AOD, and is conventionally considered as the ground truth for evaluating model-simulated (Zhai et al., 2021; Meng et al., 2021; Jin et al., 2023) or satellite-retrieved AOD (Levy et al., 2013; Wang et al., 2014a; Ridley et al., 2012; Kahn et al., 2005; Lyapustin et al., 2018)."

Satellite AOD does have higher spatial coverage, but it is also subject to uncertainties in surface reflectance properties (Remer et al., 2005; Levy et al., 2013; Kahn et al., 2005), which is avoided by ground-based measurements such as AERONET. We believe AERONET data is temporally and spatially sufficient since it provides long-term observation for more than 400 sites around the globe. Because of the low uncertainties and high amount of data, AERONET is considered the ground truth by many studies for evaluating model-simulated or satellite-retrieved AOD.

13. Lines 335-336. Who does "as a function of the parameterized surface Reff for SNA and OM" refer to specifically?

This means we plotted the discrepancy as the y-axis and the parameterized $R_{eff}$ as the x-axis (left and middle penal in Figure 6 in the manuscript). We revised lines 357-360 to make it clearer: "The left and middle panel of Figure 6 show the discrepancy between GEOS-Chem simulated AOD and AERONET AOD as a function of the parameterized surface $R_{eff}$ for SNA and OM. The simulation using the default $R_{eff}$ slightly overestimates AOD at sites with small parameterized $R_{eff}$ and underestimates AOD at sites with large parameterized $R_{eff}$."

14. Figure B4. The figure only shows the difference between AERONET and the default GEOS-Chem simulation and the difference between the new parameterization and the default simulation, which does not capture the impact of the new parameterization, so why not directly show the difference between AERONET or other observations with the new parameterization and the default simulation?

This concern can be addressed by Figure 6 in the manuscript a "Scatter plot of AERONET versus simulated AOD with the default Reff (blue dots, line, and text), and with the parameterized Reff (red dots, line, and text)."

We discuss this figure in the last paragraph in section 3.3: "The underestimates are mitigated when applying the parameterized Reff in GEOS-Chem (Figure 6, middle panel), yielding increased consistency between the measured (AERONET) AOD and simulated AOD (Figure 6, right; R2 change from 0.68 to 0.74, slope from 0.75 to 0.94)." where Figure B4 is mentioned as a supplement to illustrate the capability of our parameterization (background) to mitigate the existing biases vs. AERONET (dots) in the default scheme.

**RC2 by Referee #2**

Here the authors present work on the use of observations and global size-resolved model output to parameterize and improve representation of particle dry effective radius in bulk aerosol simulations. Data from two airborne campaigns is used to fit $R_{eff}$ to two chosen predictive covariates: total mass of SNA and OM particulates, and OM/SNA ratio. On the whole I find this work to be an interesting, well-composed, and valuable contribution to the aerosol modeling literature, and have only a few concerns and questions for the authors before publication.

While the bulk modeling is done on a relatively high resolution cubed-sphere grid, much of the parameterization relies on output from a much coarser 4x5 TOMAS simulation. While I recognize the computational time limitations inherent to high resolution global simulations, I'm a little concerned that important features may be completely washed out in such coarse output. It would be helpful to see a comparison between key metrics for the 4x5 run and a higher resolution alternative, even if just for a selected month or two.

To address this concern about resolution, We added in the last paragraph of section 2.2, lines 200-204: "For computational feasibility, a one-year global simulation is conducted with a horizontal resolution of $4° × 5°$ and 47 vertical layers from surface to 0.01 hPa. The spin-up time is 1 month. Aerosols are tracked in 15 size bins with particle diameters ranging from about 3 nm to 10 µm. We also conducted a $2° × 2.5°$ simulation for October to evaluate the sensitivity of our conclusions to the resolution of the aerosol microphysics simulation."

We also added in line 318: "Evaluation of our parameterization versus the GEOS-Chem-TOMAS simulation of $2° × 2.5°$ for October yields similar results but explains an additional 14% of the variance in simulated $R_{eff}$, providing additional evidence of the fidelity of the parameterization."

Model resolution is an interesting and important factor to consider. Our analysis showed that although some metrics such as $M_{SNAOM}$ and $R_{eff}$ vary greatly across resolutions, the relationship among them hardly changed. This is because what our parameterization reflects is the aerosol microphysics processes in the model, which are not very sensitive to model emissions or resolutions. We compared a one-month GEOS-Chem-TOMAS $4° × 5°$ simulation and a $2° × 2.5°$ simulation for October 2017. A finer resolution does provide better details, such as hotspots in boreal forests in North America and Europe, higher maximum $M_{SNAOM}$ (96.4 µg/m³ vs. 57.4 µg/m³), and higher maximum $R_{eff}$ (208 nm vs. 179 nm). There is less dust influence which makes it possible to sample 20.8% more grid boxes for parameterization.

Despite all these differences, the responses of sampled data to the parameterization are similar (Figure S-3), with $R^2$ around 0.35 and slopes around 0.9 for PBL data. The $R^2$ for total column data increased from 0.62 to 0.76 benefiting from more sampled data and a wider $R_{eff}$ range. There is an 8% increase in slope from 1.18 to 1.28, which might be suggesting that the $4° × 5°$ simulation is subject to sampling bias. But the difference is not strong enough to justify the usage of a $2° × 2.5°$ simulation.

[Figure]

Figure S-3. Agreement between parameterized $R_{eff}$ and simulated $R_{eff}$ for (left) a 4 X 5 GEOS-Chem-TOMAS simulation and (right) a 2 x 2.5 simulation.

I don't see any mention of spinup time preceding analyzed model output. Please include this modeling detail in section 2.2.

The spin-up time was 1 month for both simulations. This information has been added to section 2.2 (line 181 and lines 201-202):

"A global GCHP simulation (Eastham et al. 2018) version 13.0.0 (DOI: 10.5281/zenodo.4618180) that includes advances in performance and usability (Martin et al., 2022), is conducted on a C90 cubed-sphere grid corresponding to a horizontal resolution of about 100 km, with a spin-up time of 1 month."

"For computational feasibility, a one-year global simulation is conducted with a horizontal resolution of $4° \times 5°$ and 47 vertical layers from surface to 0.01 hPa. The spin-up time is 1 month."

One part of the parameterization process selects for locations dominated by $M_{SNAOM}$. However, if I understand Figure 3 correctly, this subset excludes a very large fraction of the surface from the calculation, and I'm unclear on the intent and consequences of this when the resulting parameterizations are applied over areas that were excluded from fitting. Is the resulting parameterization only subsequently applied to the SNA+OM fraction of particulate mass? How is this combined with other aerosol species in terms of radiative properties in the bulk simulation? The description of this process in 3.3 does not adequately cover some of these details.

Regarding the question about the intent and consequences of the sampling, we excluded grid boxes with less than 90% aerosol being SNA or OM for two reasons: First, the way that GEOS-Chem currently calculates AOD is by treating SNA, OM, sea salt, dust, and black carbon as externally mixed, using species-specific optical properties to calculate and combine AOD attributable to each species. Therefore, we are seeking a parameterization that can represent the size of SNA and OM aerosol when there is little interference from other species (when there is mostly SNA and OM). Second, the aerosol composition and size distribution measurements from aircraft campaigns are mostly SNA and OM dominated, which is chosen as the main scope of this study. Limited measurement is available for evaluating dust influence during the campaigns. It would be an interesting topic for future study when more measurement data are available.

To clarify the intent and consequences, we updated the first paragraph of section 3.2, lines 245-260 to: "Given the strong positive correlation of aerosol mass with aerosol size, we further examine this relationship globally using GEOS-Chem coupled with the TOMAS aerosol microphysics scheme. To focus on areas that are dominated by SNA and OM, we only include grid boxes with $M_{SNAOM} > 90\%$ of the aerosol mass…The top panel of Figure 3 shows the geographic distribution of annual mean surface layer dry $R_{eff}$ for grid boxes that meet the criterion."

And we updated the 5[th] paragraph of section 3.3 (about lines 338-340) to: "By design, the parameterization has little effect in regions and seasons where and when $M_{SNAOM}$ is not dominant, since the parameterization only affects $R_{eff}$ of SNA and OM."

I have some concerns over the presentation of Figure 4. First, the chosen color scheme strikes me as somewhat odd, as it uses a diverging bar centered in very light hues at

around 110 nm. This creates some (potentially unintended?) artifacts in the perception of differences between values near the center vs differences at the extremes of the displayed size range. Unless there is a compelling reason to use a diverging scheme centered at 110, I would recommend a more balanced sequential scheme. Based on the text description, it also appears that the colorbar is saturated fairly aggressively, potentially washing out model differences at the extremes. This presentation choice should be more clearly described, and the behavior outside of the saturated bounds should be discussed as needed.

We updated figure 4 with a more balanced sequential color scheme:

[Figure]

Figure S-4. Reproduction of Figure 4 with a sequential color scheme.

To address concerns about the saturated ends, we added a discussion in lines 309-314: "At the lower end of $R_{eff}$, the agreement between simulation and the parameterization can also be found in **Error! Reference source not found.**, which shows that the small $R_{eff}$ are reproduced by the parameterization. Despite the overall consistency, a few differences exist. When aerosol mass concentration is high, the parameterization tends to yield a higher $R_{eff}$ than in the GEOS-Chem-TOMAS simulation, since the adjustment using aircraft measurements led to 11% increase in $R_{eff}$."

Finally, while I recognize this parameterization as a potentially valuable addition over current defaults used in bulk aerosol schemes, I couldn't help but wonder about how the chosen parameter covariates relate to the overall mass distribution, rather than just the resulting $R_{eff}$. While a full examination of this may be unnecessary and out of scope, it would be helpful to have some context regarding the size distributions shifts surrounding the Reff differences associated with SNA+OM and OM/SNA. Are the changes in $R_{eff}$ mostly driven by shifts in the heavy tail? Are the mean changes more proportionally seen across the size distribution? A little bit more info here would be very interesting, and also suggestive of possibilities for future work stemming from this manuscript.

To provide more information about these changes, we revised the text to state in lines 256-259: "Inspection of the GEOS-Chem-TOMAS size distribution across continental regimes reveals a general tendency for the distribution to shift toward smaller sizes as $R_{eff}$ decreases and toward larger sizes as $R_{eff}$ increases, thus supporting the use of the single summary statistic of $R_{eff}$ for aerosol size."

[revised manuscript text omitted]